# Videogame exposure positively associates with selective attention in a cross-sectional sample of young children

Alexandria D. Samson[1,2,3], Christiane S. Rohr[1,4,5,6], Suhyeon Park[1,4,7], Anish Arora[1,4,7], Amanda Ip[1,4,5,6], Ryann Tansey [1,4,5,6], Tiana Comessotti[4], Sheri Madigan[1,8], Deborah Dewey[1,9], Signe Bray [1,4,5,6]*

**1** Alberta Children's Hospital Research Institute, Cumming School of Medicine, University of Calgary, Calgary, Alberta, Canada, **2** Rotman Research Institute, Baycrest Centre, Toronto, Ontario, Canada, **3** Department of Psychology, University of Toronto, Toronto, Ontario, Canada, **4** Child and Adolescent Imaging Research (CAIR) Program, Cumming School of Medicine, University of Calgary, Calgary, Alberta, Canada, **5** Hotchkiss Brain Institute, Cumming School of Medicine, University of Calgary, Calgary, Alberta, Canada, **6** Department of Radiology, Cumming School of Medicine, University of Calgary, Calgary, Alberta, Canada, **7** Faculty of Medicine and Dentistry, University of Alberta, Edmonton, Alberta, Canada, **8** Department of Psychology, University of Calgary, Calgary, Alberta, Canada, **9** Department of Community Health Sciences, Cumming School of Medicine, University of Calgary, Calgary, Alberta, Canada

* slbray@ucalgary.ca

**Data Availability Statement:** All relevant data are within the manuscript and its Supporting Information files.

## Abstract

There is growing interest in how exposure to videogames is associated with young children's development. While videogames may displace time from developmentally important activities and have been related to lower reading skills, work in older children and adolescents has suggested that experience with attention-demanding/fast-reaction games positively associates with attention and visuomotor skills. In the current study, we assessed 154 children aged 4–7 years (77 male; mean age 5.38) whose parents reported average daily weekday recreational videogame time, including information about which videogames were played. We investigated associations between videogame exposure and children's sustained, selective, and executive attention skills. We found that videogame time was significantly positively associated only with selective attention. Longitudinal studies are needed to elucidate the directional association between time spent playing recreational videogames and attention skills.

## Introduction

There is growing concern about young children's exposure to screen-media activities (SMA) [1, 2], such as television or videogames, as excessive exposure may displace time from other developmentally important activities and could impact brain, behaviour, and cognitive development [3–6]. In particular, use of videogames in young children may be increasing with the proliferation and accessibility of gaming devices [7, 8]. A 2015 study in a large UK cohort showed that ~70% of 5-year old children play video/electronic games regularly [9]. Early childhood is a particularly interesting and important period in which to examine associations

**Funding:** This work was supported by: Natural Sciences and Engineering Research Council of Canada (NSERC) Discovery Grant (DG 435787) to SB [www.nserc-crsng.gc.ca]. Natural Sciences and Engineering Research Council of Canada (NSERC) Undergraduate Student Research Award (USRA) award to AS [www.nserc-crsng.gc.ca]. Canadian Institutes of Health Research (CIHR) Project Scheme award (Project 156415) to SB [cihr-irsc.gc.ca]. Alberta Innovates Health Solutions (AIHS) Postdoctoral Award to CR [albertainnovates.ca]. The funders had no role in study design, data collection and analysis, decision to publish, or preparation of the manuscript.

**Competing interests:** The authors have declared that no competing interests exist.

between videogame use and cognition, due to rapid brain, cognitive, and behavioural development [10–13], which may confer greater susceptibility to experience-dependent plasticity.

While much screen-media literature has focused on potential negative impacts and associations (such as with attention deficit hyperactivity disorder behaviours [5, 14–16]), interactive screen-media also has the potential to support learning (e.g., language learning, history, and physical education; [17]), and in some contexts, has shown positive associations with attention and other cognitive skills [18–22]. Videogames give users practice with maintaining vigilance over long periods, ignoring distractions, and making fast motor responses. Literature in older children and adults suggests that playing videogames with an 'action' component, i.e., placing specific demands on visual-spatial attention [21], associates with enhanced perceptual skills [19, 20], visual selective attention [23] and, visuomotor skills [24].

Attention is a multi-faceted cognitive domain that has been conceptualized in a tripartite model including sustained, selective, and executive components [25–27]. All three types of attention are maturing in young children [25]. This may perhaps confer an opportunity for experience-dependent developmental plasticity such that spending time playing videogames that engage attention may associate with greater skills. Further, with a growing interest in the use of 'serious-games' (e.g., educational videogames used to enhance working memory) for cognitive therapeutic purposes [28], it is valuable to determine whether videogames that place demands on vigilance and fast-reaction played recreationally associate with cognitive benefits in young children.

The goal of the present study is to assess whether there are associations between videogame exposure and attention skills in young children. Specifically, this study examined cross-sectional associations between parent-reported weekday recreational videogame use and children's selective, sustained, and executive attention skills. Based on prior literature in young adults [23, 29], we hypothesized that time spent playing videogames would be associated with better selective attention.

## Materials and methods

### Participants

Data was collected at the Alberta Children's Hospital from 180 typically developing children between the ages of 4-7-years. Participants were recruited through community advertisements, the Healthy Infants and Children's Clinical Research Program (HICCUP) at the University of Calgary, and by reaching out to families who gave consent to be contacted during enrollment in other University of Calgary research studies. Parents provided informed written consent and children provided written assent. Study procedures were approved by the Conjoint Health Research Ethics Board of the University of Calgary. Exclusion criteria were: history of a neuro-developmental disorder, psychiatric and/or neurological diagnosis, contraindications to magnetic resonance imaging (study participation involved neuroimaging session, not reported here), or chronic medical condition. Fourteen children withdrew from the study. Twelve sibling pairs participated; one child per family was chosen at random to be included in analyses. Data was analyzed for 154 children (77 female) who had complete videogame data; however, not all 154 participants had complete data on maternal education (n = 2) and cognition outcomes (selective attention, n = 4; visual sustain attention, n = 8; auditory sustained attention, n = 5; executive attention, n = 7; IQ, n = 2). Participants were excluded from any analyses for which they had missing data.

### Procedure

Study participation involved three separate 1.5- to 2-hour visits to the Alberta Children's Hospital that typically took place within two weeks of each other. During these sessions, parents

completed questionnaires (10% were fathers) and children participated in cognitive assessments. The IQ assessments were conducted by a trained psychometrist and other assessments were conducted by a research assistant.

## Demographic questionnaire

Parents reported on maternal and paternal education and occupation. For analyses presented here, maternal education (highest degree completed) was used as an indicator of family socio-economic status (SES) and was grouped into five categories: high school diploma, technical/trade school degree, college diploma, bachelor's degree, or graduate/professional school.

## Videogame assessment

Parents completed a questionnaire [9] that asked them to estimate their child's typical weekday time spent playing videogames at home over the past two weeks. Videogames included games that were played on the internet, television, or handheld device. Daily weekday time was measured because children's weekday schedules are more consistent than weekends [30]. For each game, parents were asked to record their child's daily participation time using the following time intervals: 0 min, <30min, 30min-1hr, 1hr-3hr, and >3hr, which were later converted to a score from 0 (0 min) to 4 (>3hr). Parents were asked to report the name of each game, and the device it was played on. To estimate total videogame time, time spent playing each game reported was summed.

## General cognition

To characterize the sample, intelligence was assessed using the Wechsler Preschool and Primary Scale of Intelligence– 4[th] Edition CDN (WPPSI-IV; [31]). Full Scale IQ (mean = 108, standard deviation = 13) was estimated from the scores on the following sub-tests of the WPPSI-IV: Block Design, Information, Matrix Reasoning, Bug Search, Picture Memory, and Similarities. These sub-tests were chosen as representative measures for each of the five WPPSI-IV sub-domains (visual spatial, fluid reasoning, working memory, processing speed, verbal comprehension). A test in each sub-domain is needed to calculate a Full Scale IQ for children 4 to 7 years of age.

## Attention assessments

The Early Childhood Attention Battery (ECAB; [25]) was used to assess attention. In previous work, split-half reliability was found to be good (r = 0.75) as was validity relative to the Test of Everyday Attention for Children [26] (Pearson r = 0.765; [25]). Children completed eight sub-tests of the ECAB, four of which were included in this study to assess three components of attention: selective attention, sustained attention (visual sustained attention and auditory sustained attention), and executive attention. All of the sub-tests, except the selective attention task, were administered via a Dell laptop computer (screen size 31 cm by 17.5 cm), at a 35–50 cm viewing distance. The auditory components of the tasks were played through a set of external speakers. All computerized tasks included a practice trial, which was repeated if the child demonstrated they did not understand the instructions.

**i. Selective attention.** Selective attention was measured using the ECAB visual search task. Children were given one 60 s trial to point to targets (18 red apples, each 1.5 x 1.5 cm) among distractors (162 white apples and red strawberries, each 1.5 x 1.5 cm) on a laminated letter-sized search sheet. The experimenter marked items with an erasable marker as children pointed to them. This test was scored by summing correctly identified targets.

**ii. Visual sustained attention.**    For the ECAB visual sustained attention measure, a series of pictures was presented on a computer screen (200 ms presentations with an inter-stimulus interval [ISI] of 1800 ms). The child was asked to verbally say "yes", "animal", or the name of the animal when a picture of an animal appeared on the screen. Thirty targets and 120 non-targets (familiar everyday items, e.g., car or bike) were presented over a 5-minute session. All pictures depicted animals or objects with monosyllabic names. Children were prompted to pay attention if they missed four consecutive targets. The score was the sum of the correct animal responses minus any errors (responses to non-targets) and prompts.

**iii. Auditory sustained attention.**    Children also completed a similar auditory sustained attention task from the ECAB in which a continuous stream of words (monosyllabic animal target words and familiar item non-target words) were presented (average duration 650 ms, ISI 1,350 ms) over 5-minute task duration. The child was again asked to say "yes", "animal", or the name of the animal when they heard an animal name. The measure was scored as the number of correct responses minus any errors and prompts.

**iv. Executive attention.**    Executive attention was assessed using the ECAB's child-appropriate adaptation of a Wisconsin Card Sorting Test [32]. Children had to deduce which kind of balloon a teddy bear preferred. Each trial showed two balloons that varied in colour and shape. In stage one, the teddy bear liked one colour of balloon; in stage two, the preferred colour changed; and in stage three, the bear preferred balloons of a particular shape. Children received visual feedback on whether their choice was correct, but no other information was given. A total of 20 possible trials could be given for each stage, with six consecutive correct trials required for a pass to the next stage. If a child failed a stage, the test was discontinued. The task was scored as the total number of stages (three total) that were successfully completed.

## Data analysis

All analyses were completed in R version 3.6.2. Mann-Whitney Wilcoxon tests were used to assess sex differences in age. Fisher's exact test was used to assess sex differences in maternal education, videogame time, and videogame use. We examined whether demographic data (age, sex, and maternal education) were associated with the predictor (time spent playing videogames) or the outcome (attention measures) variables. Sex and age were correlated with many of the predictors and outcome measures; therefore, they were used as covariates in all models. Maternal education, as a marker of SES, was controlled for in follow-up analyses. As the videogame time measure was non-linear, we used partial Spearman correlations to assess relationships between outcomes and videogame time. Inferences were drawn at p < 0.05 Bonferroni corrected for multiple comparisons across attention measures; specifically, p-values were divided by four.

## Exploratory analysis based on videogame content

Some literature on videogames has distinguished 'action' from 'non-action' games with action games being games that involve rapid pacing, switching attention between vigilance across the visual field to monitor for potential threats, and focusing in to accomplish a specific task and often include the control of an avatar (e.g. first-person shooter; [18]). While first-person shooter videogames are not commonly used by young children, we conducted an exploratory analysis modeled on the action/non-action distinction stratifying games based on a rough estimation of the cognitive demand, into 'fast-reaction games' or 'slow-reaction games'. Fast-reaction games were those that included challenges such as hand-eye coordination and time pressure to make a response, while the latter mainly involved games with an educational component such as learning math, the alphabet, or social skills. Games were classified by two

independent raters using http://igdb.com and other online sources including videos about the games on youtube.com. There was high inter-rater reliability between videogame classifications with 97% agreement but, in the case of a disagreement, a third rater acted as a tiebreaker. For categorical analyses, children were grouped as 'non-gamers' who did not play any videogames, 'slow-reaction gamers' who played only slow-reaction games, and 'fast-reaction gamers' who played fast-reaction games (with or without also playing slow-reaction games). Follow-up exploratory analyses used analysis of covariance (ANCOVA) models to examine associations between attention measures and gamer-type (non-gamer, slow-reaction gamer, fast-reaction gamer).

## Results

### Sample demographics

Participant demographics are shown in Table 1. We note that the distribution of maternal education in our sample was skewed towards the upper end of our scale. As seen in Table 1, boys and girls did not differ in terms of age or maternal education. In our sample, boys scored lower than girls on IQ (Table 1). We did not find significant sex differences on attention measures (Table 2).

### Videogame use and associations with demographics

Videogame play was common in this sample but not ubiquitous, with more than half of children spending some time playing videogames (Table 1). The median of videogame time was 0–30min per weekday; among children who played games, the median was 30min -1h. Time spent playing videogames did not differ by sex (Table 1). Age was significantly positively

**Table 1. Demographics, general cognition, time spent playing videogames, and videogame use.**

|  | All | Female | Male | p |
|---|---|---|---|---|
| N | 154 | 77 | 77 | |
| Age (median [IQR]) | 5.43 [4.52, 6.13] | 5.36 [4.46, 5.98] | 5.61 [4.60, 6.17] | 0.26 |
| IQ (mean (SD)) | 108.72 (12.57) | 111.29 (10.87) | 106.09 (13.68) | **0.01** |
| Maternal Education (%) | | | | 0.99 |
| High school | 7 (4.6) | 3 (3.9) | 4 (5.3) | |
| Trade/technical | 5 (3.9) | 2 (2.6) | 3 (4.0) | |
| College diploma | 31 (20.3) | 16 (20.8) | 15 (20.0) | |
| Bachelor's degree | 69 (45.1) | 35 (45.4) | 34 (45.4) | |
| Graduate/professional | 40 (26.1) | 21 (27.3) | 19 (25.3) | |
| Videogame Time (%) | | | | 0.23 |
| 0min | 68 (44.1) | 36 (46.8) | 32 (41.5) | |
| < 30min | 26 (16.9) | 10 (13.0) | 16 (20.8) | |
| 30min - 1h | 22 (14.3) | 14 (18.2) | 8 (10.4) | |
| 1- 3h | 16 (10.4) | 5 (6.5) | 11 (14.3) | |
| > 3h | 22 (14.3) | 12 (15.6) | 10 (13.0) | |
| Videogame Use | | | | 0.63 |
| None (%) | 68 (44.1) | 36 (46.8) | 32 (41.6) | |
| Some (%) | 86 (55.8) | 41 (53.2) | 45 (58.4) | |

Scores are presented for the entire sample and for male and female participants separately. For non-normally distributed scores, median and interquartile ranges are given. Wilcoxon test was used for age. A two-sample t-test was used for IQ. Fisher's exact tests were used for maternal education, videogame time, and videogame use.

**Table 2. Attention measures.**

|  | All | n | Female | Male | p |
|---|---|---|---|---|---|
| Executive attention (median [IQR]) | 34.70 (12.88) | 147 | 32 [26, 41] | 32 [26, 37] | 0.462 |
| Selective attention (median [IQR]) | 13.57 (3.22) | 150 | 14 [12, 16] | 14 [11, 16] | 0.813 |
| Visual sustained (median [IQR]) | 23.44 (6.30) | 146 | 26 [21, 28] | 25 [21, 28] | 0.979 |
| Auditory sustained (median [IQR]) | 20.81 (7.27) | 150 | 22 [19, 27] | 22 [17, 26] | 0.207 |

Scores are presented for the entire sample and for male and female participants separately. For non-normally distributed scores, median and interquartile ranges are given. A two-sample t-test was used for IQ, other scores were compared using Kruskal-Wallis test.

correlated with time spent playing videogames ($r_s$ (152) = 0.20, p < 0.05). General cognition, measured by Full Scale IQ, was associated at trend-level with time spent-playing videogames when controlling for age and sex ($r_s$ (150) = -0.14, p = 0.08). Maternal education was significantly associated with time spent on videogames ($r_s$ (150) = -0.16, p < 0.05).

### Associations between videogame use and attention skills

Time spent playing videogames was positively associated with selective attention ($r_s$ (148) = 0.20, p < 0.05) when controlling for age and sex (Fig 1). This association survived Bonferroni correction and remained significant when additionally controlling for maternal education ($r_s$ (146) = 0.22, p < 0.01). Additionally, when an outlier was removed from the selective attention scores the association remained, again controlling for age, sex, and maternal education ($r_s$ (145) = 0.24, p < 0.01). Spearman correlations revealed no significant association with the other attention measures when controlling for age and sex (visual sustained: $r_s$ (144) = 0.046, p > 0.05; auditory sustained: $r_s$ (147) = 0.00076, p > 0.05; executive: $r_s$ (145) = -0.023, p > 0.05).

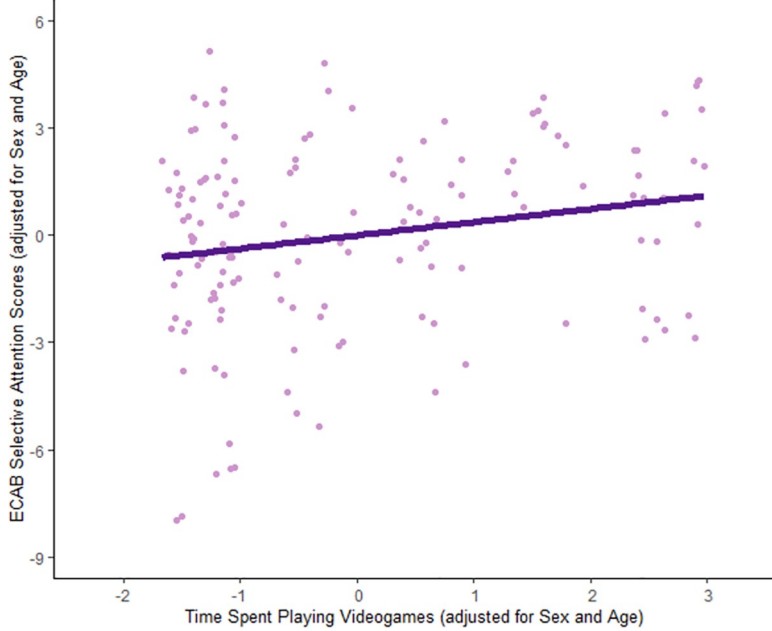

**Fig 1. Association between time spent playing videogames and selective attention scores.** Partial Spearman correlation showed a significant positive association between time spent playing videogames and selective attentions scores when controlling for age and sex ($r_s$ (148) = 0.20, p < 0.05). ECAB = Early Childhood Attention Battery.

**Table 3. Gamer type in this sample.**

|  | All | Female | Male | p |
|---|---|---|---|---|
| Gamer Type (%) |  |  |  | 0.20 |
| Non-gamer | 68 (44.2) | 36 (46.7) | 32 (41.5) |  |
| Slow-reaction gamer | 36 (23.3) | 21 (27.3) | 15 (19.5) |  |
| Fast-reaction gamer | 50 (32.5) | 20 (26.0) | 30 (39.0) |  |

Scores are presented for the entire sample and for male and female participants separately. Fisher's exact test was used.

### Exploratory analyses based on videogame content

There were 226 different videogames reported that were classified as fast-reaction (n = 72) or slow-reaction games (n = 154). For example, *Subway Surfers* [33] was considered a fast-reaction game while *My Little Pony*: *Friendship Gardens* [34] was considered a slow-reaction game.

Fast-reaction, slow-reaction, and non-gamer type did not differ by sex (Table 3). Age was significantly associated with gamer type ($F_{(1,152)}$ = 8.7, $p < 0.01$, $np^2$ = 0.054), with fast-reaction gamers ($t_{(100.6)}$ = 2.9, $p < 0.01$; mean = 5.60) older than non-gamers (mean = 5.16). However, fast-reaction and slow-reaction gamers (mean = 5.48) did not significantly differ in age ($t_{(77.3)}$ = 0.64, $p > 0.05$) nor did slow-reaction and non-gamers ($t_{(68.8)}$ = 1.93, $p > 0.05$). Maternal education showed a trend level association with gamer type ($F_{(1, 150)}$ = 3.09, $p = 0.08$).

Gamer type was associated with selective attention scores in an ANCOVA model that controlled for age and sex ($F_{(1, 146)}$ = 6.34, $p < 0.05$, $np^2$ = 0.042); however this association did not survive Bonferroni correction. There was no association between gamer type and other attention measures when controlling for age and sex (visual sustained: $F_{(1, 142)}$ = 2.06, $p > 0.05$, $np^2$ = 0.014; auditory sustained: $F_{(1, 145)}$ = 0.24, $p > 0.05$, $np^2$ = 0.0095; executive: $F_{(1, 143)}$ = 0.027, $p > 0.05$, $np^2$ = 0.00019).

## Discussion

This study investigated whether time spent playing recreational videogames was associated with children's attention skills. We found that more than half of the children in our sample regularly played videogames and that use was more frequent in older children. We found a positive association between time spent playing videogames and selective attention skills but not visual sustained, auditory sustained, or executive attention skills. Importantly, longitudinal studies are needed to determine whether children with greater selective attention skills play videogames more frequently because they are more predisposed to be good at them, or whether there is a practice-dependent effect.

Our findings suggest that there is a potential 'advantage' in terms of selective attention skills in children who spend more time playing videogames. These findings are consistent with previous literature in adults. For example, action videogame experts (≥7 hours of videogame play per week) compared to non-gamers (≤1 hour of videogame play per week) showed enhanced task-switching abilities, a selective attention skill, [35] and comparable results have been demonstrated elsewhere [36, 37]. Furthermore, the ability to filter irrelevant information, another selective attention skill, was shown to be enhanced in action gamers (≥ 5 hours of videogame play per week) compared to non-gamers (≤1 hour of videogame play per week) in conjunction with gamers displaying different neural activity, suggested to be associated with greater attentional control [29]. However, dissimilar to these adult studies, we note that our effect of

selective attention was associated with videogame duration rather than with a particular type of videogame.

We note that this potential 'advantage' of recreational videogame duration did not generalize to other attention skills assessed here. Studies in adults have shown associations between playing action videogames and executive attention skills, such as enhanced executive functioning [38] and executive control [39]. Studies in young children, on the other hand, have shown mixed results with regards to time spent on videogames and executive functioning skills. In line with our finding of no association between videogame use and executive attention, Jusienė et al. [40] found that SMA exposure, including TV, computer, smartphone, and tablet use, in preschool-aged children did not associate with executive function. However, a screen intervention study that assessed young children's (2 and 3 years old) executive functioning before and after a screen activity, demonstrated that playing an educational app (similar to slow-reaction games here) had no significant adverse effects on children's executive functioning and even improved scores on one executive function task compared to children who passively watched an educational television show or cartoon [41].

We found no association with sustained attention here. Previous work investigating sustained attention and videogaming in adolescents found that across a sustained attention task, action gamers (≥7 hours of videogame play per week) showed a greater performance decline relative to non-action gamers (≤1 hour of videogame play per week) [42]. These authors speculate that because of the fast-paced environment that action gamers are used to, this may cause them more difficulty when trying to complete relatively mundane sustained attention tasks. No studies to our knowledge have shown positive associations.

In terms of associations between time spent playing recreational videogames and demographic variables, we found a positive association with age, a negative association with maternal education, and no association with sex. Similarly, fast-reaction gamers were significantly older than non-gamers ($p < 0.05$) and mothers of fast-reaction gamers on average had lower educational attainment relative to non-gamers ($p = 0.05$). Although we expected to find that boys spent more time playing videogames than girls, we did not see this or a sex-specific effect for gamer-type. It is possible that a distinction between sexes in videogame use may emerge later in development [43] as games developed for young children tend to focus on appealing to both sexes (e.g., see S1 Table). On the other hand, it could be that sex differences in children's videogame use is diminishing as the use of handheld games becomes more common.

One of the motivations for examining recreational videogame use in relation to attention skills is motivated by the use of 'serious-games' for cognitive rehabilitation. This is an active, though sometimes controversial [44], area of research, and recent preliminary work has shown positive effects of cognitive videogame training on visual working memory and selective attention in young children (6–12 years) with Autism Spectrum Disorder [45]. Furthermore, various studies [46] have illustrated the benefits in using videogames to improve symptoms in children with attention deficit hyperactivity disorder [46, 47] and to improve reading abilities and attentional skills in children with dyslexia [48]. Thus, videogame play has the potential to be a therapeutic tool in clinical settings. Our study suggests that the use of recreational videogames should be considered in the context of cognitive rehabilitation, as a potential confound and, given the positive association with selective attention found here, support integrating features of recreational games into therapeutic games.

Strengths of the present study include the use of direct assessment of children's attention skills in an early childhood sample and analyses based on videogame duration and characteristics. However, several study limitations should also be noted. Maternal education, used here as a proxy for SES, was skewed toward the upper end of our scale, limiting the generalizability of our findings. The relatively small sample size may have limited our power to detect small

effects as significant. Our data are cross-sectional and therefore cannot be used to assess the causal effects of videogames on cognition or attention. We did not collect information about the social context of videogames or parenting style [43]. As noted above, our criteria for 'fast-reaction' games differed from that used to classify videogames played by older children and adults (i.e., 'action' games [18]). Finally, parent- reports likely underestimate actual use of videogames and mobile devices in young children [49] as well, only recreational games were included in this study therefore games played in school, at a friend's house, or elsewhere were not taken into account. Future work should incorporate objective measurements for more accurate videogame usage including time-use diaries and passive sensing [50]

## Conclusions

Videogaming is increasingly common in young children and to support recommendations on SMA use for families it is important to understand whether videogaming associates with children's attention skills and whether associations are positive or negative. Focusing on a tripartite model of attention, we found a significant positive association between time spent playing recreational videogames and selective attention. Importantly, our cross-sectional findings cannot be used to infer causality or directionality of these associations, and future longitudinal or interventional studies are needed to determine whether playing videogames in early childhood influences attention development.

## Supporting information

**S1 Table. List of videogames played by participants.**
(DOCX)

**S1 Dataset. Raw data.**
(XLSX)

## Acknowledgments

The authors gratefully acknowledge study support from Sarah Vinette, Ivy Cho, Amy Webber, and Kari Parsons. We also thank all of the families who took the time to participate in this research.

## Author Contributions

**Conceptualization:** Deborah Dewey, Signe Bray.

**Data curation:** Alexandria D. Samson, Christiane S. Rohr, Suhyeon Park, Anish Arora, Amanda Ip.

**Formal analysis:** Alexandria D. Samson, Suhyeon Park, Anish Arora, Amanda Ip, Tiana Comessotti, Sheri Madigan, Signe Bray.

**Investigation:** Amanda Ip, Ryann Tansey, Tiana Comessotti.

**Methodology:** Christiane S. Rohr, Signe Bray.

**Project administration:** Amanda Ip, Signe Bray.

**Supervision:** Christiane S. Rohr, Amanda Ip, Sheri Madigan, Deborah Dewey, Signe Bray.

**Writing – original draft:** Alexandria D. Samson, Christiane S. Rohr, Sheri Madigan, Signe Bray.

**Writing – review & editing:** Alexandria D. Samson, Christiane S. Rohr, Suhyeon Park, Anish Arora, Amanda Ip, Ryann Tansey, Tiana Comessotti, Sheri Madigan, Deborah Dewey, Signe Bray.

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
