## [Decision Letter · Decision Letter 0]

12 Feb 2021

PONE-D-20-30582

Screen-games and their association with cognition and behaviour in young children

PLOS ONE

Dear Dr. Bray,

Thank you for submitting your manuscript to PLOS ONE. After careful consideration, we feel that it has merit but does not fully meet PLOS ONE’s publication criteria as it currently stands. Therefore, we invite you to submit a revised version of the manuscript that addresses the points raised during the review process.

Firt of all, we would like to apologize for the delay in sending the reviewers´s comments to the authors. This time was specially hard for finding available reviewers, as the COVID-19 is still dramatically affecting our lifes in different senses.

Based on the comments of the reviewers and my own opinion, I recommend a major revision of your manuscript, paying special attention at the reviewers´ comments concerning background, methods and results. Also, a revision and editing of the english language is required. 

We look forward to receiving your revised manuscript.

Kind regards,

Trinidad Garcia, PhD

Academic Editor

PLOS ONE

Journal Requirements:

3.We note that the grant information you provided in the ‘Funding Information’ and ‘Financial Disclosure’ sections do not match.

Reviewers' comments:

Reviewer's Responses to Questions

**Comments to the Author**

1. Is the manuscript technically sound, and do the data support the conclusions?

Reviewer #1: Partly

Reviewer #2: Partly

2. Has the statistical analysis been performed appropriately and rigorously? 

Reviewer #1: No

Reviewer #2: Yes

3. Have the authors made all data underlying the findings in their manuscript fully available?

Reviewer #1: Yes

Reviewer #2: Yes

4. Is the manuscript presented in an intelligible fashion and written in standard English?

Reviewer #1: No

Reviewer #2: Yes

5. Review Comments to the Author

Reviewer #1: The authors examine the linkage between young children's time spent playing digital games, diverse indices of their attentional abilities including sustained and selective attention, and ADHD and ASD symptoms.

The reasons for studying these linkages is not well-justified. In fact, the authors' claims about the relative dearth of information about young children's digital game exposure as linked to cognitive skills reflect limited consultation of developmental research that would show otherwise. Specifically, the authors are urged to consult the work emanating from the labs of, for example, Heather Kirkorian and Ellen Wartella. Consideration of the characteristics of the games and apps that young children use and the cognitive and academic skills that may be promoted through their play also is available in research done by those within the developmental psychology and educational technology communities. Reasons for linking game play to ASD symptoms is not well grounded in the literature aside from these symptoms being shown more frequently among males than girls; the latter of whom are presumed to spend more time playing digital games than girls. This presumption alone, particularly among young children, warrants greater substantiation than the authors provide.

The methods section was far too limited in the information provided. For example, which parent completed the surveys? Did parents have full knowledge of children's game activities outside the home, given that many preschools may not include digital game play as part of the academic curriculum. Similarly, were games included if they were presented as part of a website that included activities for children? Distinctions between what qualified as action v. non-action game were fuzzy. What exactly would qualify as a non-action game? In looking at Table 2, it seems as if those games that were more educational in nature were more inclined to be labeled as non-action games. What was the inter-rater reliability for making distinctions between the two types of games as none is reported. What was the reason for including particular subtests of the WPPSI-IV? What exactly did the attention assessments as modeled on ECAB entail? The authors should be willing to provide at least a brief description. How many trials in the selective attention task? What was the basis for the sustained attention task used in the study and who developed it? What were the criteria that needed to be met for successful completion of forms in the VMI task? For the behavioral assessments, were the parents the reporters or were independent assessments made as the authors make mention on page 4 of using a multi-informant approach. In fact, the authors later note in their discussion section that a strength of the study was this approach. Thus, specific mention of these informants is warranted.

The results section was somewhat baffling to me as I failed to understand why participants with missing or outlier data were still included in the analyses. How were the missing data points handled within the data set? Similarly, why were those participants who exceeded clinical cut-offs on the AQC still included in the study?

Overall, I am not sure what story is told by the minimal significant findings or why the study was done. The distinction between action and non-action games seems particularly blunt and the multi-informant approach is not at all obvious.

Reviewer #2: This paper investigates the association between video-games and cognitive measurements and hyperactivity and autistic traits measured by questionnaires filled by parents in preschool children. This is a very well-written and sounded paper that I recommand for publication with minor revisions that I detail here below:

-The title is not very accurate : there is no association between screen-games and cognition and the association with behavior is in fact an association with parents' perception of their children's behavior. I would suggest to put the result : video-games were not associated with cognitive abilities but with behavior reported by parents.

- Define what is screen-media activities and what is screen-games. Why the authors do not use the term video-games?

-The introduction can be better structured. Here is my suggestion : begin with defintiions, then prevalence and then the effects of games on cognition. Then you can develop in a separate section the effect on hyperactivity and autistic traits. This would help formalize better hypotheses about the association between hyperactivty and autistic symptoms. In fact, I did not fully understand the association between video-games and autistic symptoms.

-In your hypothesis "We hypothesized that time spent playing screen-games would be positively associated with ADHD and ASD symptoms and that playing attentionally demanding games would be associated with better selective attention and visuomotor integration", how can you reconcile a positive effect on cognition and a negative effect on behavior?

-Hyperactivity and autistic traits were studied through questionnaires by parents and these were the only significant correlations with screen-games. This was not discussed in the discussion. How much reliable are parents' answers ? This study shows a correlation between games and parents' perception of the level of activity of their children and some of their personaity traits. This hihglight should have implications on the title, the abstract and the discussion.

-The table with demographics is good and could contain the percentages so the authors do not repeat the figures in the paragraph below.

-The table with the distinction between action games and non action games is not very interesting. You may give an example.

-There is a sentence in the discussion saying that this study is correlational and other studies need to study the causal effects. I agree and in my view this is not sufficiantly developped in the discussion especially when showing the positive correlations between games and hyperactivity and autistic traits. My concern is that this may mislead some readers or can be misinterpreted by journalists and parents.

6. PLOS authors have the option to publish the peer review history of their article (what does this mean?). If published, this will include your full peer review and any attached files.

Reviewer #1: No

Reviewer #2: **Yes: **Rana Esseily

---

## [Author Response · Author response to Decision Letter 0]

30 Apr 2021

Reviewer #1

The authors examine the linkage between young children's time spent playing digital games, diverse indices of their attentional abilities including sustained and selective attention, and ADHD and ASD symptoms.

1. The reasons for studying these linkages is not well-justified. In fact, the authors' claims about the relative dearth of information about young children's digital game exposure as linked to cognitive skills reflect limited consultation of developmental research that would show otherwise. Specifically, the authors are urged to consult the work emanating from the labs of, for example, Heather Kirkorian and Ellen Wartella. Consideration of the characteristics of the games and apps that young children use and the cognitive and academic skills that may be promoted through their play also is available in research done by those within the developmental psychology and educational technology communities. 

We agree that the Introduction and justification for the study could benefit from a more in-depth description of the prior literature. We have now substantially revised the Introduction to focus more directly on the main novel contribution of our work, which is direct assessment of attention skills in young children, and also referenced the important work by Wartella and colleagues [14, 19] and other development psychology research on related topics (i.e., Dale, Bavelier & Green (2020) [10]; McNeill et al., (2019) [18]; Jusiene et al. (2020) [21]). Please see the Introduction section for all of the changes made (Pages 3 – 5, Lines 26 -81). 

Further, work by Kirkorian and colleagues [46] was added to the Discussion (Lines 434 – 435)

2. Reasons for linking game play to ASD symptoms is not well grounded in the literature aside from these symptoms being shown more frequently among males than girls; the latter of whom are presumed to spend more time playing digital games than girls. This presumption alone, particularly among young children, warrants greater substantiation than the authors provide.

We agree that the rationale for examining ASD symptoms was under-developed and have decided to remove measures of ASD symptoms and behaviours from the manuscript. 

3. The methods section was far too limited in the information provided. For example, which parent completed the surveys? Did parents have full knowledge of children's game activities outside the home, given that many preschools may not include digital game play as part of the academic curriculum. Similarly, were games included if they were presented as part of a website that included activities for children? 

We thank the reviewer for noting details that should be added to the Methods section. These questions helped us to improve the clarity of this section. 

- Which parent completed the surveys? 

We have added this to the Procedure of the Methods section: “During these sessions, parents completed questionnaires (10% were fathers) and children participated in cognitive assessments.” (Lines 108 –110)

- Did parents have full knowledge of children's game activities outside the home, given that many preschools may not include digital game play as part of the academic curriculum? 

Great question, “Parents completed a questionnaire [17] that asked them to estimate their child’s typical weekday time spent playing videogames at home over the past two weeks.” (Lines 122-123) 

- Similarly, were games included if they were presented as part of a website that included activities for children? 

Parents provided information about devices used to play videogames and names of the videogames played. Based on Google searches of the games that were named by the parents, some were internet games. This has now been added to the manuscript for clarification: “Videogames included games that were played on the internet, television, or handheld device.”(Line 123 - 124) 

4. Distinctions between what qualified as action v. non-action game were fuzzy. What exactly would qualify as a non-action game? In looking at Table 2, it seems as if those games that were more educational in nature were more inclined to be labeled as non-action games. What was the inter-rater reliability for making distinctions between the two types of games as none is reported. 

We agree that the distinction between action and non-action games was not clear therefore we now classify the games as attention-demanding and non-attention-demanding games. We believe this classification better captures the way in which we rated the games. For example, as the reviewer mentions the non-attention-demanding games are typically educational in nature thus require less demand of attention. 

“The cognitive psychology literature typically distinguishes ‘action’ from ‘non-action games with action games being games that involve rapid pacing, switching attention between vigilance across the visual field to monitor for potential threats, and focusing in to accomplish a specific task and often include the control of an avatar (e.g. first-person shooter; [10]). However, these types of games are not commonly used by young children therefore we decided to categorize the games as ‘attention-demanding’ and ‘non-attention-demanding’ games. The former involved games with challenges such as fast reaction time, hand-eye coordination, and time pressure to make a response while the latter, involved games with an educational component such as learning math, the alphabet, or social skills.” (Lines 139 – 143). 

Inter-rater reliability was reported: “There was high inter-rater reliability between videogame classifications with 97% agreement but, in the case of a disagreement, a third rater acted as a tiebreaker.” (Lines 145 – 147) 

5. What was the reason for including particular subtests of the WPPSI-IV? 

Great question. We added an explanation to the methods section: “These sub-tests were chosen as representative measures for each of the five WPPSI-IV sub-domains (visual spatial, fluid reasoning, working memory, processing speed, verbal comprehension).” (Lines 160 – 162)

6. What exactly did the attention assessments as modeled on ECAB entail? The authors should be willing to provide at least a brief description. How many trials in the selective attention task? What was the basis for the sustained attention task used in the study and who developed it? 

Thank you for the questions, we have elaborated on the ECAB in the Methods section. Specifically, we have clarified how all four of the cognitive attention measures were derived from the ECAB. The ECAB measurement is broadly discussed followed by detailed explanations of the four sub-tests that were used in the study (Lines 166 – 211). 

7. What were the criteria that needed to be met for successful completion of forms in the VMI task? 

We decided to streamline our paper to focus on attention and VMI was removed from the analysis. We also note that because this measure was added after data collection began and therefore was only available for a smaller sample (n = 129), making comparisons to other outcomes reported more complicated. 

8. For the behavioral assessments, were the parents the reporters or were independent assessments made as the authors make mention on page 4 of using a multi-informant approach. In fact, the authors later note in their discussion section that a strength of the study was this approach. Thus, specific mention of these informants is warranted.

We have decided to use the term multi-method instead of multi-informant to clarify that different types of methods were used to collect the data for this study; parent-reports of ADHD behaviour traits and videogame data in addition to direct assessments of children’s attention and general cognition. We have made this more explicit within the Methods section as well as in the Introduction (e.g. Introduction section: “As a step towards understanding the potential impact of videogames on attention and cognition in early childhood, we describe a cross-sectional study examining associations between parent-reported videogame use and assessments of children’s attention skills.” (Lines 68 – 70). 

9. The results section was somewhat baffling to me as I failed to understand why participants with missing or outlier data were still included in the analyses. How were the missing data points handled within the data set? Similarly, why were those participants who exceeded clinical cut-offs on the AQC still included in the study?

We have now clarified that participants with missing data on a particular assessment were not included in the analyses that included that measure as a predictor or outcome. 

For example, if a participant did not have a selective attention score, they were not included in analyses when selective attention was being assessed. Additionally, the one selective attention outlier was removed from the model of videogame play and selective attention scores in order to test whether the effect remained significant (Line 285-287). 

We have now added a clarification to the Participants section: “Data was analyzed for 154 children (77 female) who had complete videogame data; however, not all 154 participants had complete data on maternal education (n = 2), cognitive outcomes (selective attention, n = 4; visual sustain attention, n = 8; auditory sustained attention, n = 5; executive attention, n = 7; IQ, n = 2), and behavioural traits (SNAPI, n = 2; SNAPH, n = 3; SNAPC, n = 3). Participants were excluded from any analyses for which they had missing data.”(Lines 98 – 103)

Analyses involving the AQC have now been removed from the manuscript.

10. Overall, I am not sure what story is told by the minimal significant findings or why the study was done. The distinction between action and non-action games seems particularly blunt and the multi-informant approach is not at all obvious.

As the reviewer suggested and as mentioned above, we added rationale to the Introduction including emphasis on literature of videogames in older children as well as videogames in younger children (see response to #1) to highlight the rationale for conducting this study. We have also clarified the distinction between game characteristics (see response to #4) and clarified what we meant by multi-informant approach (see response to #8). 

Reviewer #2 

This paper investigates the association between video-games and cognitive measurements and hyperactivity and autistic traits measured by questionnaires filled by parents in preschool children. This is a very well-written and sounded paper that I recommend for publication with minor revisions that I detail here below:

1. The title is not very accurate: there is no association between screen-games and cognition and the association with behavior is in fact an association with parents' perception of their children's behavior. I would suggest to put the result: video-games were not associated with cognitive abilities but with behavior reported by parents.

The reviewer raises an important consideration therefore we decided to change the title of the manuscript to: Videogame play associates with selective attention skills and hyperactivity in early childhood

Additionally, we took the reviewers suggestion to clarify the results of the study (i.e., Conclusion section: “We found a significant positive association between time spent playing recreational videogames and selective attention as well as a significant positive association between time spent playing recreational videogames and parent-rated hyperactivity behaviours. Importantly, our cross-sectional findings cannot be used to infer causality or directionality of these associations and future longitudinal or interventional studies are needed to determine whether playing videogames in early childhood influences cognitive or behavioral development.” (Lines 442 - 448). 

2. Define what screen-media activities is and what is screen-games. Why the authors do not use the term video-games?

We appreciate the suggestion and we decided to use videogames, instead of screen-games, throughout the manuscript. 

Additionally, examples of screen-media activities were provided: “There is growing concern about young children’s exposure to screen-media activities (SMAs) [1] such as television or videogames,…” (Lines 27 – 28). 

3. The introduction can be better structured. Here is my suggestion: begin with definitions, then prevalence and then the effects of games on cognition. Then you can develop in a separate section the effect on hyperactivity and autistic traits. This would help formalize better hypotheses about the association between hyperactivity and autistic symptoms. In fact, I did not fully understand the association between video-games and autistic symptoms.

As mentioned in the response to Reviewer 1 (response #1), the Introduction was thoroughly revised to more clearly justify the study rationale in light of existing literature. Further, as noted above, analyses of ASD traits have been removed from the manuscript.

4. In your hypothesis "We hypothesized that time spent playing screen-games would be positively associated with ADHD and ASD symptoms and that playing attentionally demanding games would be associated with better selective attention and visuomotor integration", how can you reconcile a positive effect on cognition and a negative effect on behavior?

We agree that these hypotheses may sound counterintuitive; however, directly assessed cognitive skills and parent-reported behaviour traits do not always correlate. In the ADHD literature, there is variation in the extent and nature of cognitive challenges reported. In context of our study hypotheses, we considered that there could be a possibility that a child who scores high on parent-rated hyperactive traits and frequent videogame usage, could also demonstrate high selective attention abilities through extensive practice from frequent videogaming. 

We have now added a sentence to justify the reason for our hypothesis as well as cited other resources that have found similar results: “It may be counterintuitive to hypothesize different direction of effects when considering cognitive or behavioural measures, but these hypotheses are supported by previous literature [8, 24]. Further, children in our sample did not have a diagnosis of ADHD and the literature on cognitive differences in ADHD is mixed, with few studies examining cognitive differences specific to inattentive or hyperactive ADHD traits [25].” (Lines 76 – 81)

5. Hyperactivity and autistic traits were studied through questionnaires by parents and these were the only significant correlations with screen-games. This was not discussed in the discussion. How reliable are parents' answers? This study shows a correlation between games and parents' perception of the level of activity of their children and some of their personality traits. This highlight should have implications on the title, the abstract and the discussion.

Many great points are made here. We have made sure to be more explicit in mentioning that the behaviour outcomes are based on parent reports in the Title, Abstract, and Discussion (e.g., Abstract section: “We found that videogame time was significantly positively associated with directly assessed selective attention and parent-rated hyperactivity scores, but not parent-rated inattention or other directly assessed attention skills.” (Lines 11 – 13). 

Also, we have added the parent-reports as a limitation to the study in the Discussion of the manuscript: “Finally, parent- reports likely underestimate actual use of videogames and mobile devices in young children [45]. Future work should incorporate objective measurements for more accurate videogame usage including time-use diaries and passive sensing [46] in addition to, teacher reports of children’s ADHD behaviours to corroborate the parent reports [47].” (Lines 432– 436)

6. The table with demographics is good and could contain the percentages so the authors do not repeat the figures in the paragraph below.

Following this suggestion we eliminated the percentages in the text. Now all percentages can be found in Table 1 (Line 250) 

7. The table with the distinction between action games and non action games is not very interesting. You may give an example.

The Table with the list of action and non-action games was removed. As per the response to Reviewer 1 (response #4), the games are now classified as attention-demanding and non-attention-demanding. An example of an attention-demanding game and a non-attention-demanding game were provided in the text instead: “There were 226 different videogames reported that were classified as attention-demanding (n = 72) or non-attention-demanding (n = 154). For example, Subway Surfers [31] was considered an attention-demanding game while My Little Pony: Friendship Gardens [32] was considered a non-attention-demanding game.” (Lines 262 – 265)

8. There is a sentence in the discussion saying that this study is correlational and other studies need to study the causal effects. I agree and in my view this is not sufficiently developed in the discussion especially when showing the positive correlations between games and hyperactivity and autistic traits. My concern is that this may mislead some readers or can be misinterpreted by journalists and parents.

We understand the concern, and have made sure to be clear when explaining that this study is correlational not causal whenever necessary through out the manuscript. For example, we acknowledge the correlational nature of this work in the new Title, Introduction, Results, and Discussion sections to more accurately reflect the results of the study.

---

## [Decision Letter · Decision Letter 1]

5 Jul 2021

PONE-D-20-30582R1

Videogame play cross-sectionally associates with selective attention skills and hyperactivity traits in young children

PLOS ONE

Dear Dr. Bray,

Thank you for submitting your manuscript to PLOS ONE. After careful consideration, we feel that it has merit but does not fully meet PLOS ONE’s publication criteria as it currently stands. Therefore, we invite you to submit a revised version of the manuscript that addresses the points raised during the review process.

After deeply considering the comments and suggestions made by the reviewers (which are very divergent), we consider that a new revision on the manuscript should be made before it can be considered for publication. Specially relevant are the comments from Reviewer 1, who still expressess important concerns regarding the current study. Please try to address these comments in a new version of your manuscript.

We look forward to receiving your revised manuscript.

Kind regards,

Trinidad Garcia, PhD

Academic Editor

PLOS ONE

Reviewers' comments:

Reviewer's Responses to Questions

**Comments to the Author**

1. If the authors have adequately addressed your comments raised in a previous round of review and you feel that this manuscript is now acceptable for publication, you may indicate that here to bypass the “Comments to the Author” section, enter your conflict of interest statement in the “Confidential to Editor” section, and submit your "Accept" recommendation.

Reviewer #1: (No Response)

Reviewer #2: All comments have been addressed

2. Is the manuscript technically sound, and do the data support the conclusions?

Reviewer #1: No

Reviewer #2: Yes

3. Has the statistical analysis been performed appropriately and rigorously? 

Reviewer #1: N/A

Reviewer #2: Yes

4. Have the authors made all data underlying the findings in their manuscript fully available?

Reviewer #1: (No Response)

Reviewer #2: Yes

5. Is the manuscript presented in an intelligible fashion and written in standard English?

Reviewer #1: Yes

Reviewer #2: Yes

6. Review Comments to the Author

Reviewer #1: I remain unconvinced that the changes here help to better substantiate the need for the study. For example, I fail to understand how the study shared here provides nuanced information to address mixed findings in the data concerning the benefits or liabilities of SMAs which sometimes seem to refer to videogames and sometimes to a larger category of screen devices. I am no clearer on what constitutes attention-demanding v. non-attention demanding which is likely the eyes of the player rather than the investigators. Without far greater substantiation, I do not buy into the assumption that action games are attentionally demanding and educational games not, especially for younger children.

I do not understand why the examination of ADHD "traits" and why the display of those traits at a non-clinical level is necessarily bad. Again, why only parent informants here warrants explanation. I remain unclear about what is meant by attention skills, which is not a monolithic construct or why selective attention in particular is targeted within the hypothesis. The methods section does make it clearer that different forms of attention are examined. However, reasons for their investigation remain unspecified in the study rationale.

I was further confused as to why maternal education was examined and how game play was assessed. For example, if children are enrolled in day care or some preschool program, might it be the case that children play digital games in those settings? For that matter, might they play games on apps with their friends or a sibling? I ask as I find it unusual that so many children would be characterized as "no-gamers."

Regardless, the distinction between types of games children play is no clearer to me than in the earlier version of the study. I can appreciate the authors' goals to analyze data that they have about children's game play and individual differences but a better reason for doing so needs to be apparent as do far clearer measures and characterizations of the games played for readers to identify the story being shared by the findings.

Reviewer #2: The authors have now answered all of my questions and the article has been revised accordingly. I believe that the paper can be published now.

7. PLOS authors have the option to publish the peer review history of their article (what does this mean?). If published, this will include your full peer review and any attached files.

Reviewer #1: No

Reviewer #2: **Yes: **Rana Esseily

---

## [Author Response · Author response to Decision Letter 1]

27 Aug 2021

Response to Reviewers

We thank the reviewers again for their valuable feedback on our manuscript, which we have integrated into this revised submission. We have incorporated most of the suggestions made by Reviewer #1. The reviewers questions are bolded, text from the manuscript are “italicized and in quotations”, with new additions/revisions to the manuscript in blue and the corresponding lines are highlighted in yellow. Please see below for a point-by-point response. 

Altogether, we feel this new revision of the manuscript encompasses the key contributions of our study and is therefore more focused and concise. Again, we would like express our gratitude towards the reviewers for their positive contribution to this manuscript. 

Reviewer #1: 

1. I remain unconvinced that the changes here help to better substantiate the need for the study. For example, I fail to understand how the study shared here provides nuanced information to address mixed findings in the data concerning the benefits or liabilities of SMAs which sometimes seem to refer to videogames and sometimes to a larger category of screen devices. 

In light of this and comment #3, we have substantially revised the manuscript beginning with the Introduction, to focus on the main contributions of our study, which is associating recreational videogame exposure with attention skills in an early childhood sample. We acknowledge the confusion and inconsistency in using both screen-media activities (SMA) and videogames therefore, we now mention SMA once in the Introduction to illustrate that videogaming is a type of SMA: “There is growing concern about young children’s exposure to screen-media activities (SMA) [1,2], such as television or videogames, as excessive exposure may displace time from other developmentally important activities and could impact brain, behaviour, and cognitive development [3–6].” (Lines 23 – 26) and once in the Conclusion to suggest that our findings can support videogaming guidelines for young children thus, help guide children SMA guidelines as a whole: “Videogaming is increasingly common in young children and to support recommendations on SMA use for families it is important to understand whether videogaming associates with children’s attention skills and whether associations are positive or negative.” (Lines 357 - 359). 

2. I am no clearer on what constitutes attention-demanding v. non-attention demanding which is likely the eyes of the player rather than the investigators. Without far greater substantiation, I do not buy into the assumption that action games are attentionally demanding and educational games not, especially for younger children.

Our goal was to draw parallels to literature in older children and adults suggesting that there is a specific benefit of action games in terms of attention skills. However, we acknowledge that we used internet descriptions of games for classification and did not do a detailed content analysis of each game, nor have our definitions of different types of games been validated. However, we felt that the detail we collected in terms of games children play was valuable and worth exploring. We have now therefore moved the analysis based on game content to sections that are explicitly labeled ‘exploratory’ on page 10 in the Methods section (Lines 174 – 193) and pages 13 and 14 in the Results section (Lines 254 – 271).

Moreover, to better reflect the videogames that were played by the children the games are now referred to as ‘fast-reaction games’ and ‘slow-reaction games’ rather than ‘attention demanding’ and ‘non-attention demanding’ (Lines 182 – 185). 

3. I do not understand why the examination of ADHD "traits" and why the display of those traits at a non-clinical level is necessarily bad. Again, why only parent informants here warrants explanation. 

This part of the study was a replication of prior work with less informative data. Given the limited information collected here on parenting and family environment, we have now removed analyses related to attention-deficit hyperactivity disorder (ADHD) traits from the manuscript to focus our study on attention skills more specifically. 

4. I remain unclear about what is meant by attention skills, which is not a monolithic construct or why selective attention in particular is targeted within the hypothesis. The methods section does make it clearer that different forms of attention are examined. However, reasons for their investigation remain unspecified in the study rationale.

In terms of study rationale, we have now clarified through the Introduction that because children spend a lot of time playing videogames, which place demands on attention skills, there is an opportunity for experience-dependent plasticity and the potential for games to associate with attention skills as a result. Specifically, we hypothesize that selective attention, the ability to focus and ignore distractions, may be enhanced in children who spend more time playing videogames.

We now more clearly articulate in the Introduction that this study used a tripartite model of attention and the rationale for looking at attention in relation to videogames: “Attention is a multi-faceted construct that has been conceptualized in a tripartite model including sustained, selective, and executive components [25-27]. All three components are maturing in young children [25], perhaps conferring an opportunity for experience-dependent developmental plasticity with videogame use. Spending time playing videogames that engage attention may therefore associate with greater skills in these areas. Further, with a growing interest in the use of ‘serious-games’ (e.g., educational videogames used to enhance working memory) for cognitive therapeutic purposes [28], it is valuable to determine whether videogames that place demands on vigilance and fast-reaction played recreationally are associated with cognitive benefits in young children.” (Lines 41 – 49). 

We now more clearly articulate our reasoning for this hypothesis in the Introduction: “Based on prior literature in young adults [23,29], we hypothesized that time spent playing videogames would be associated with better selective attention.” (Lines 53 – 55).

Additionally, in the Methods section we specified that the Early Childhood Attention Battery (ECAB) was used to assess three components of attention in line with the tripartite model. “Children completed eight sub-tests of the ECAB, four of which were included in this study to assess the three components of attention: selective attention, sustained attention (visual sustained attention and auditory sustained attention), and executive attention.” (Lines 115 – 118). 

5. I was further confused as to why maternal education was examined and how game play was assessed. For example, if children are enrolled in day care or some preschool program, might it be the case that children play digital games in those settings? For that matter, might they play games on apps with their friends or a sibling? I ask as I find it unusual that so many children would be characterized as "no-gamers." Regardless, the distinction between types of games children play is no clearer to me than in the earlier version of the study. 

As noted in the manuscript (Lines 85- 89) maternal education was used as proxy for socioeconomic status: “For analyses presented here, maternal education (highest degree completed) was used as an indicator of family socioeconomic status (SES) and was grouped into five categories: high school diploma, technical/trade school degree, college diploma, bachelor’s degree, or graduate/professional school.”

We are now clearer that we assess only recreational game play in the home. For example, in the Introduction: “The goal of the present study is to assess whether there are associations between videogame exposure and attention skills in young children. Specifically, this study examined associations between parent-reported weekday recreational videogame use and direct assessments of children’s selective, sustained, and executive attention skills.” (Lines 50 – 53). As well as in the Conclusion: “Finally, parent- reports likely underestimate actual use of videogames and mobile devices in young children [35] as well as, only recreational games were included in this study therefore games played in school, at a friend’s house, or elsewhere were not taken into account.” (Lines 349 – 352). 

As noted above, we have removed the attention-demanding game distinction from the main analyses to an exploratory analysis as well as renamed the games to better characterize the games played by the children (please see response to comment #2). 

6. I can appreciate the authors' goals to analyze data that they have about children's game play and individual differences but a better reason for doing so needs to be apparent as do far clearer measures and characterizations of the games played for readers to identify the story being shared by the findings.

We have taken this feedback to heart and as noted above have further streamlined the paper to focus on the key contributions: weekday time spent playing recreational videogames and selective attention skills in typically developing young children. As mentioned above in comment #5, we have attempted to be more clear in introducing the three measures of attention that are being assessed according to the tripartite model of attention. Additionally, as mentioned in the manuscript, a validated battery of attention was used to measure the three types of attention which is more than what other studies have done. Furthermore, as there are no standardized and validated parent reports or other measures for videogame use in young children we would have used them but as this is a rapidly evolving field validated measures have yet to be established. For these reasons, we made the analyses regarding videogame content and their characterization exploratory given the lack of prior literature guiding the distinction between types of games. Overall, our approach mirrors and extends on previously published research and provides insights into a very timely topic, at a crucial time in children’s cognitive development.

Reviewer #2: 

The authors have now answered all of my questions and the article has been revised accordingly. I believe that the paper can be published now.

---

## [Editor Report · Decision Letter 2]

14 Sep 2021

Videogame exposure positively associates with selective attention in a cross-sectional sample of young children

PONE-D-20-30582R2

Dear Dr. Bray,

We’re pleased to inform you that your manuscript has been judged scientifically suitable for publication and will be formally accepted for publication once it meets all outstanding technical requirements.

Kind regards,

Trinidad Garcia, PhD

Academic Editor

PLOS ONE

Additional Editor Comments (optional):

Many thanks for sending a revised version of the manuscript. Based on its current state, the Editor considers it can be published. 
---

## [Editor Report · Acceptance letter]

17 Sep 2021

PONE-D-20-30582R2 

Videogame exposure positively associates with selective attention in a cross-sectional sample of young children 

Dear Dr. Bray:

I'm pleased to inform you that your manuscript has been deemed suitable for publication in PLOS ONE. Congratulations! Your manuscript is now with our production department. 

Kind regards, 

on behalf of

Dr. Trinidad Garcia 

Academic Editor

PLOS ONE